# QTL Mapping and Candidate Gene Identifying for N, P, and K Use Efficiency at the Maturity Stages in Wheat

**DOI:** 10.3390/genes14061168

**Published:** 2023-05-27

**Authors:** Xu Han, Mingxia Zhang, Minggang Gao, Yuanyuan Yuan, Yapei Yuan, Guizhi Zhang, Yanrong An, Ying Guo, Fanmei Kong, Sishen Li

**Affiliations:** 1National Key Laboratory of Wheat Improvement, College of Agronomy, Shandong Agricultural University, Tai’an 271018, China; hanxu92@foxmail.com (X.H.); zhangmingxia0506@163.com (M.Z.); chgaoming@126.com (M.G.); happyxinhai20047@163.com (Y.Y.); guizhi0715@163.com (G.Z.); anyanrong2002@163.com (Y.A.); guoying729@126.com (Y.G.); 2Key Laboratory of Biochemistry and Molecular Biology, College of Biological and Agricultural Engineering, Weifang University, Weifang 261061, China; 3Jinan Academy of Agricultural Science, Jinan 250316, China; 4National Engineering Laboratory for Efficient Utilization of Soil and Fertilizer Resources, College of Resources and Environment, Shandong Agricultural University, Tai’an 271018, China; yyp325325@163.com; 5Institute of Industrial Crops, Shandong Academy of Agricultural Sciences, Jinan 250108, China

**Keywords:** wheat, mineral use efficiency (MUE), mineral uptake efficiency (MUpE), mineral utilization efficiency (MUtE), quantitative trait locus (QTL), recombinant-inbredline (RIL)

## Abstract

Nitrogen (N), phosphorus (P), and potassium (K) are the three most important mineral nutrients for crop growth and development. We previously constructed a genetic map of unigenes (UG-Map) based on their physical positions using a RIL population derived from the cross of “TN18 × LM6” (TL-RILs). In this study, a total of 18 traits related to mineral use efficiency (MUE) of N/P/K were investigated under three growing seasons using TL-RILs. A total of 54 stable QTLs were detected, distributed across 19 chromosomes except for 3A and 5B. There were 50 QTLs associated with only one trait, and the other four QTLs were associated with two traits. A total of 73 candidate genes for stable QTLs were identified. Of these, 50 candidate genes were annotated in Chinese Spring (CS) RefSeq v1.1. The average number of candidate genes per QTL was 1.35, with 45 QTLs containing only one candidate gene and nine QTLs containing two or more candidate genes. The candidate gene *TraesCS6D02G132100* (*TaPTR* gene) for *QGnc-6D-3306* belongs to the *NPF* (*NRT1/PTR*) gene family. We speculate that the *TaPTR* gene should regulate the GNC trait.

## 1. Introduction

Wheat (*Triticum aestivum* L.) is a major crop around the world. It is a staple food for 30% of the world’s population [1] and accounts for approximately one-fifth of the total calories consumed by humans [2]. Nitrogen (N), phosphorus (P), and potassium (K) are the most essential and important nutrient elements for plant growth and development [3]. N fertilizer is the main input used to boost grain yield and grain protein content, and agricultural production relies on synthetic N fertilizer to increase productivity [4]. P is an important component of plant biomolecules such as DNA, RNA, ADP, ATP, and NDPH. This makes it the second-most essential nutrient after N for plant development and crop production. As an essential macronutrient, K is involved in most biochemical reactions in plants, such as stomatal regulation, osmoregulation, protein synthesis, carbohydrate metabolism, and enzyme activation. In these processes, K can regulate the rate of generative reactions [5]. In recent decades, global fertilizer application has resulted in a significantly increased application of N, P, and K [6]. In an economic sense, fertilizer application has exceeded the optimal application rate, resulting in a massive waste of resources and economic losses [7]. Reliance on chemical fertilizers is clearly not a long-term solution, as both their production and application cause pollution and threaten environmental sustainability.

Mineral use efficiency (MUE) consists of two components: mineral uptake efficiency (MUpE), the ability of plants to remove minerals from the soil, and mineral utilization efficiency (MUtE), the ability of plants to use minerals to produce grain [8,9,10]. In wheat, the MUE of N, P, and K varies greatly among genotypes by growth stage, and the analysis of the MUE genetic basis is of great value [11,12,13,14,15,16]. The development of new high-yielding crop varieties under relatively low N/P/K environments is an effective way to improve MUE [17,18,19,20].

The traits associated with MUE are quantitative traits. Quantitative trait loci (QTL) analysis provides an efficient way to breakdown complex traits into loci and study their relative effects on specific traits [21]. In wheat, a large number of QTLs related to MUE have been mapped in low and high nutrient environments [22,23,24,25,26,27,28]. For example, Zhang et al. [29] detected 121 and 130 QTLs for N use efficiency traits under hydroponic and field treatments, respectively. Yang et al. [30] detected twelve new QTLs influencing root and biomass-related traits for N uptake under three hydroponic N levels. Su et al. [31] identified seven QTLs related to P utilization efficiency under high and low P treatments. Yuan et al. [14] detected 55 and 68 QTLs related to P use efficiency under hydroponic and field treatments, respectively. Gong et al. [32] detected a total of 127 QTLs related to K use efficiency under 10 different N and P concentrations. Safdar et al. [33] identified two QTLs related to P use efficiency in a genome-wide association study and found that six genes were highly expressed in pyruvate metabolism and the TCA cycle under low phosphorus treatment.

We previously constructed a set of recombinant inbred lines (RILs) derived from the “TN18 × LM6” cross. The objective of the present study was to perform QTL analysis for MUE traits in field trials of RILs at different treatments of N, P, and K and then to find relatively stable QTLs and candidate genes.

## 2. Materials and Methods

### 2.1. Plant Materials

The population used for QTL analysis in this study consisted of a set of RILs derived from the cross of “TN18 × LM6” (TL-RILs, 184 lines) [14,29]. The female parent, TN18, is a cultivar developed by our group that was approved by the Crop Variety Approval Committee (CVAC) of Shandong Province in 2008. TN18 was planted on approximately 300,000 hectares per year in the Huang-Huai Winter Wheat Region, China. The male parent, LM6, is an elite wheat line whose mother is a sister line of the famous cultivar Jimai 22. The two parents showed obvious differences in MUE, with the LM6 having a higher MUE than the TN18 as a whole. A total of 184 lines of RILs were randomly selected from the original 305 lines to be studied.

### 2.2. Experimental Design

We constructed eight nutrient plots of 110 m^2^ (10 × 11 m) in the experimental station of Shandong Agricultural University for mineral nutrient element experiments. The plots were divided by 1.5-meter-deep cement brick walls. The soil structure was maintained as a natural field with fertile soil [8]. Mineral nutrients were depleted by growing wheat and maize each year until nutrient levels met trial requirements. Three field trials were conducted during the 2012–2013, 2013–2014, and 2014–2015 growing seasons.

Four treatments were designed in the three trials: CK (target grain yield: 9000 kg/ha) and low N (LN), low P (LP), and low K (LK) (target grain yield: 6000 kg/ha for each corresponding element) [14,29]. All P_2_O_5_ and K_2_O and 60% of the N were applied before sowing, and 40% of the N was applied at the stem elongation stage. Each trial was a complete block design with two replications in two nutrient plots. Each line of TL-RILs was sown in two rows, with 1 m in length and 25 cm between rows. Twenty seeds for one row were sown in 2012–2013 and 2013–2014, and 40 seeds for one row were sown in 2014–2015. Seeds were sown on 15 October, and plants were harvested on 13–15 June.

### 2.3. Trait Measurement

A total of 18 MUE traits were investigated (Table 1), including 12 MUpE traits (GNC, GPC, GKC, StNC, StPC, StKC, GNA, GPA, GKA, StNA, StPA, and StKA) and six MUtE traits (GNUE, GPUE, GKUE, StNUE, StPUE, and StPUE). We randomly harvested the above-ground parts of 10 plants of each line in each replicate, and all traits were determined using mixed samples from the same treatment. Of the investigated data, GNC, StNC, GNA, StNA, GNUE, StNUE, GPC, StPC, GPA, StPA, GPUE, and StPUE from 2012–2013 and 2013–2014 were previously determined [14,29].

### 2.4. Data Analysis

Analysis of variance (ANOVA) was calculated using SPSS 18.0 software (SPSS Inc., Chicago, IL, USA). Since the traits were measured with mixed samples, a two-factor ANOVA was used. The genotypes and treatments were considered two factors using the average values of three growing seasons. All factors involved were considered sources of random effects. The generalized heritability (*h_B_*^2^) was estimated by the following formula: *h_B_*^2^ = *σ_g_*^2^/(*σ_g_*^2^ + *σ_e_*^2^), where *σ_g_*^2^ is the genotype variance and *σ_e_*^2^ is the total error variance.

### 2.5. QTL Detection and Candidate Gene Identification

The genetic map of unigenes (UG-Map) for the TL-RILs was used for QTL analysis. Each line of the TL-RILs was sequenced using RNA-Seq technology. The clean reads were mapped to the Chinese Spring (CS) RefSeq v1.1. A total of 31,445 polymorphic sub-unigenes were obtained, which represented 27,983 unigenes. The sub-unigenes were used to construct the UG-Map based on their physical positions [34]. Three software programs were used for QTL analysis: Windows QTL Cartographer 2.5 (http://statgen.ncsu.edu/qtlcart/WQTLCart.htm (accessed on 1 August 2019), IciMapping 4.1 (http://www.isbreeding.net/ accessed on 28 January 2019), and MapQTL 6.0 (https://www.kyazma.nl/index.php/mc.MapQTL/ accessed on 1 August 2019) [35]. IciMapping 4.1 adopts the complete interval mapping method. Both the Windows QTL Cartographer 2.5 and MapQTL 6.0 software used the composite interval mapping method. For the three software programs, the LOD value was 3.0 and the step size was 0.5 cM. For a single software program analyzing data from the same treatment, we defined a stable QTL when it was detected over the AV + 1 environments. Finally, we employed the stable QTLs that were detected in at least two of the three software programs.

The unigene(s) covered or rounded (not exceeding the middle distance between two unigenes) by the peak position region of a QTL were regarded as the candidate gene(s) of the corresponding QTLs.

## 3. Results

### 3.1. Phenotypic Variation and Heritability

For the parents of the RIL population, TN18 and LM6 showed obvious differences in most investigated traits (Appendix A). For the RIL population, a wide range of variations existed. Transgressive segregation was observed for all of the 144 trait-treatment combinations (including AV). All of the 18 investigated traits in each treatment showed continuous distributions, which indicates their quantitative nature. The *h_B_*^2^ for the investigated traits ranged from 61.75% (StKC) to 88.33% (GNC) (Appendix A).

### 3.2. Major Characteristics of the Stable QTLs

Using Windows QTL Mapper 2.5, IciMapping 4.1, and MapQTL 6.0 software, a total of 54 stable QTLs were detected for all 18 traits, distributed across 19 chromosomes except for 3A and 5B (Table 2 and Appendix A, Figure 1). Among them, 11 QTLs were detected in all three software programs. Twenty-two QTLs showed positive additive effects with the female parent TN18 increasing the QTL effects, whereas 32 QTLs showed negative additive effects with the male parent LM6 increasing the QTL effects. There were 50 QTLs associated with only one trait, and the other four QTLs were associated with two traits.

For the N/P/K concentration traits, 27 stable QTLs were detected (Table 2, Figure 1). Among them, 8, 6, and 3 QTLs were detected for GNC, GPC, and GKC, respectively, distributed across 11 chromosomes: 1B, 1D, 2A, 2B, 3B, 3D, 5D, 6B, 6D, 7A, and 7B. A number of 5, 2, and 3 QTLs were detected for StNC, StPC, and StKC, respectively, distributed across 10 chromosomes: 1A, 1B, 1D, 2A, 2B, 2D, 4B, 4D, 6A, and 7D.

For the N/P/K accumulation traits, a number of 10 stable QTLs were found (Table 2, Figure 1). Among them, 1, 1, and 3 QTLs were found for GNA, GPA, and GKA, respectively, distributed across 5 chromosomes: 1A, 3D, 4A, 6A, and 7B. A number of 3, 2, and 0 QTLs were found for StNA, StPA, and StKA, respectively, and were distributed across four chromosomes, 2A, 5D, 6A, and 7A.

For the MUtE traits, 13 stable QTLs were detected (Table 2, Figure 1). Among them, 1, 2, and 2 QTLs were found for GNUE, GPUE, and GKUE, respectively, distributed across five chromosomes 1A, 2B, 4D, 5A, and 6B. An amount of 1, 2, and 5 QTLs were found for StNUE, StPUE, and StKUE, respectively, distributed across 7 chromosomes: 2B, 3B, 4A, 4B, 5A, 6B, and 6D.

Four stable QTLs were associated with two traits (Table 2, Figure 1). Of these, *QStka/Stkc-2A-12963* and *QGpa/Gpc-5A-12924* were associated only with the MUpE traits. *QGpa/Stpue-2B-18259* and *QGka/Stpue-4B-1222* were associated with both the MUpE and MUtE traits.

### 3.3. Candidate Genes for Stable QTLs

A total of 73 candidate genes (including ncRNAs) for the 54 stable QTLs were identified (Table 2, Figure 1) that were covered or rounded by the regions of the peak positions of the corresponding stable QTLs. Among these candidate genes, 50 were annotated in Chinese Spring (CS) RefSeq v1.1 [2], eight were annotated in TL-RILs, and 15 were ncRNAs. The average number of candidate genes per QTL was 1.35 (73/54) with 45 QTLs (83.3%) containing only one candidate gene, seven QTLs (13.0%) containing two candidate genes, and two QTLs (3.7%) containing 3–8 candidate genes.

For the N/P/K concentration traits, 31 candidate genes were identified, with 23 genes annotated in RefSeq v1.1, one gene annotated in TL-RILs, and seven ncRNAs (Table 2, Figure 1). Of these, five (*TraesCS5D02G261500*, *TraesCS6D02G132100*, *TraesCS7A02G-364700*, *TraesCS7B02G406300*, and *TraesCS7B02G406400*), three (*TraesCS5D02G459900*, *TraesCS-6B02G012600,* and *TraesCS7A02G476700*), and three (*TraesCS1D02G024700*, *TraesCS2B02G564200,* and *TraesCS6D02G115000*) high confidence (HC) genes in RefSeq v1.1 for GNC, GPC, and GKC were found, respectively. For StNC, StPC, and StKC, two (*TraesCS2A02G213508* and *TraesCS2D02G124400*), one (*TraesCS2B02G131100*), and three (*TraesCS4D02G229500, TraesCS6A02G063300,* and *TraesCS7D02G532100*) HC genes were identified, respectively.

For the N/P/K accumulation traits, 11 candidate genes were identified, with four genes annotated in RefSeq v1.1, three genes annotated in TL-RILs, and four ncRNAs (Table 2, Figure 1). Of these, only two HC genes were obtained: *TraesCS5D02G295200* for StNA and *TraesCS6A02G372600* for StPA.

For the MUtE traits, 16 candidate genes were identified, with 12 genes annotated in RefSeq v1.1, two genes annotated in TL-RILs, and two ncRNAs (Table 2, Figure 1). Of these, two HC genes for GKUE, *TraesCS4D02G195400* and *TraesCS5A02G383800*, were identified. For StNUE, StPUE, and StPUE, two (*TraesCS4B02G033800* and *TraesCS4B02G-034500*), one (*TraesCS6B02G440300*), and four (*TraesCS3B02G008600*, *TraesCS5A02G-555500*, *TraesCS6B02G133900,* and *TraesCS6D02G214800*) HC genes were found. 

A total of fifteen candidate genes from four QTLs were associated with two traits (Table 2, Figure 1). For *QGpa/Gpc-5A-12924* and *QGpa/Stpue-2B-18259*, *TraesCS5A02G556400* and *TraesCS2B02G609800* were identified, respectively. For *QGka/Stpue-4B-1222*, 11 candidate genes were identified, included seven genes annotated in RefSeq v1.1, two genes annotated in TL-RILs, and two ncRNAs. This QTL involved five HC genes: *TraesCS4B02G-047600*, *TraesCS4B02G047700*, *TraesCS4B02G047900*, *TraesCS4B02G049300,* and *TraesCS4B-02G049400*. For *QStka/Stkc-2A-12963*, two LC genes were identified.

## 4. Discussion

The QTL locations were carried out mainly using the genetic map of DNA markers, and a great number of QTLs for MUE traits of N/P/K were identified [14,22,25,26,27,28,29,30,31,32,33]. Using a genetic map of high-density DNA markers, a QTL could cover dozens of genes [36]. Therefore, it is difficult to identify highly reliable candidate genes for corresponding gene cloning. In this study, we used UG-Map for the RNA sequencing of the RIL population [34], which can directly determine the candidate genes for the QTLs. We identified 73 candidate genes from 54 QTLs of MUE traits, with an average of 1.35 candidate genes per QTL. This should provide convenience for the cloning of related genes and the genetic improvement of wheat breeding programs. In addition, the QTLs and their candidate genes in this study were identified according to the physical positions of the reference genome. It is difficult to accurately compare QTLs obtained based on molecular markers.

Because the cloned genes and their functions for MUE were not reported, the relationship between our candidate genes and MUE needs further verification. An example with phenotypic function is the candidate gene *TraesCS6D02G132100* from *QGnc-6D-3306* for GNC. The *TraesCS6D02G132100* gene is annotated as a peptide transporter (*TaPTR* gene). *NRT1* and *PTR* belong to the same gene family *NPF* (*NRT1/PTR*, nitrate transporter 1/peptide transporter) in terms of sequence homology, i.e., proteins that mediate the transmembrane transport of substances such as nitrate and small molecular peptides of 2–3 amino acids [37]. Therefore, we speculate that the *TraesCS6D02G132100* gene should regulate the GNC trait. 

*TraesCS6A02G372600* was the candidate gene for *QStpa-6A-10015* for StPA. This gene is annotated as a phosphate regulon sensor protein (*TaPhoR* gene). The PhoR protein can act as a protease in the inducible high-affinity phosphate transport system (Pst), which is composed of Pho regulators that accomplish phosphate transport behavior via transmembrane signaling [38]. Therefore, we speculate that the *TraesCS6A02G372600* gene may regulate the StPA trait in wheat.

The *TraesCS6B02G012600* gene was the candidate gene for *QGpc-6B-361* for GPC. This gene is annotated as an F-box family protein (*TaPFB* gene). In *Arabidopsis*, *FBX2* is a protein containing WD40 and F-box structural domains that can be used to recognize specific ubiquitination targets. For example, under phosphorus-sufficient conditions, the *fbx2* mutant has higher levels of phosphoenolpyruvate carbolicase kinase 1 (PPCK1) transcripts and significantly higher root hair numbers and total intracellular phosphorus content than the wild type [39]. Therefore, we speculate that the *TraesCS2D02G124400* gene may associate with GPC in wheat.

Using the QTL location of UG-Map for TL-RILs, we previously cloned two genes, the *XIP* (*Xylanase Inhibitor Protein*) gene, which regulates the SDS-sedimentation value and dough stability time [40], and the *TaDHL* (*ATP-dependent DNA helicase*) gene, which regulates the plant height [41]. These results further showed that the candidate genes in this study are favorable for corresponding gene cloning.

For TL-RILs, the male parent TN18 is derived from the cross “Laizhou137 × Yannong 19” and the female parent LM6 is derived from the cross “924402 × Lvmai13”. The strain 924,402 is one parent of the cultivated variety “Jimai22”. Of these parents, Yannong19 and Jimai22 are the authorized varieties with great area and the core parents in wheat breeding programs in China. Moreover, the parent of TN18 is also a core parent in China. Hundreds of authorized varieties derive from Yannong19, Jimai22, and TN18. Therefore, the QTLs and their candidate genes in this study may be widely identified in the varieties of China.

## Figures and Tables

**Figure 1 genes-14-01168-f001:**
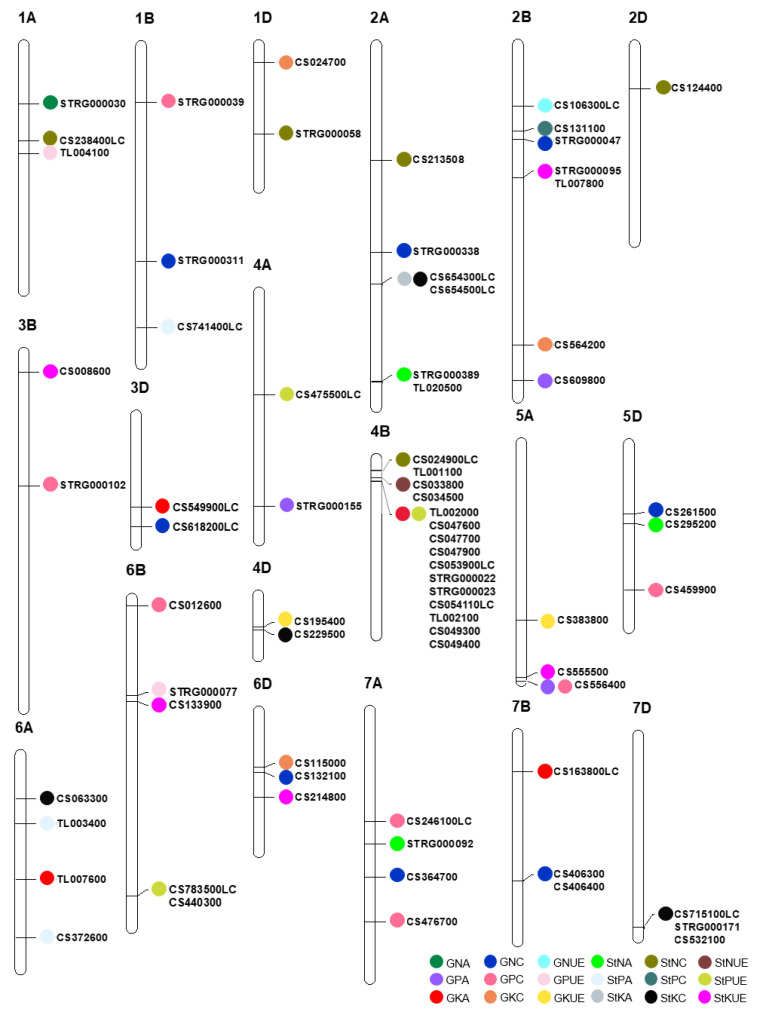
Locations of the 54 stable QTLs and their 73 candidate genes using the TL-RILs. *CS* and *TL* are abbreviations of the gene names “*TraesCSxx02G*”and “*TraesTLxx02G*”. 1A~ 7D are the names of the 21 chromosomes.

**Table 1 genes-14-01168-t001:** Summary of investigated traits and their measurement methods.

Abbreviation	Name	Unit	Measurement Method
GNC	Grain N concentration	mg·g^−1^	Using an NC analyzer (KDY-9820, Tongrunyuan Ltd., Beijing, China)
GPC	Grain P concentration	mg·g^−1^	Using a sequential plasma spectrometer (ICPS-7500; Shimadzu Corp., Kyoto, Japan)
GKC	Grain K concentration	mg·g^−1^	Using a sequential plasma spectrometer (ICPS-7500; Shimadzu Corp., Kyoto, Japan)
StNC	Straw N concentration	mg·g^−1^	Using an NC analyzer (KDY-9820, Tongrunyuan Ltd., Beijing, China)
StPC	Straw P concentration	mg·g^−1^	Using a sequential plasma spectrometer (ICPS-7500; Shimadzu Corp., Kyoto, Japan)
StKC	Straw K concentration	mg·g^−1^	Using a sequential plasma spectrometer (ICPS-7500; Shimadzu Corp., Kyoto, Japan)
GNA	Grain N accumulation	g·(m^2^)^−1^	GNC × GW
GPA	Grain P accumulation	g·(m^2^)^−1^	GPC × GW
GKA	Grain K accumulation	g·(m^2^)^−1^	GKC × GW
StNA	Straw N accumulation	g·(m^2^)^−1^	StNC × StW
StPA	Straw P accumulation	g·(m^2^)^−1^	StPC × StW
StKA	Straw K accumulation	g·(m^2^)^−1^	StKC × StW
GNUE	Grain N use efficiency	kg^2^ GW·g^−1^ GNA	GW^2^/GNA
GPUE	Grain P use efficiency	kg^2^ GW·g^−1^ GPA	GW^2^/GPA
GKUE	Grain K use efficiency	kg^2^ GW·g^−1^ GKA	GW^2^/GKA
StNUE	Straw N use efficiency	kg^2^ StW·g^−1^ StNA	StW^2^/StNA
StPUE	Straw P use efficiency	kg^2^ StW·g^−1^ StPA	StW^2^/StPA
StKUE	Straw K use efficiency	kg^2^ StW·g^−1^ StKA	StW^2^/StKA

GW—grain weight; StW—straw weight.

**Table 2 genes-14-01168-t002:** Stable QTLs and their candidate genes in TL-RILs. “*TraesTLxx02G……*” are the genes that are annotated in TL-RILs; “*STRG……*” are the noncoding RNAs (ncRNAs).

Trait	Chrom	QTL	Gene	Annotation in RefSeq v1.1
GNC	1B	*QGnc-1B-11704*	*STRG_1B.000311*	-
GNC	2A	*QGnc-2A-11241*	*STRG_2A.000338*	-
GNC	2B	*QGnc-2B-5134*	*STRG_2B.000047*	-
GNC	3D	*QGnc-3D-6047*	*TraesCS3D02G618200LC*	Glutathione S-transferase T3
GNC	5D	*QGnc-5D-3809*	*TraesCS5D02G261500*	F-box protein
GNC	6D	*QGnc-6D-3306*	*TraesCS6D02G132100*	Peptide transporter
GNC	7A	*QGnc-7A-9048*	*TraesCS7A02G364700*	Histone-lysine N-methyltransferase
GNC	7B	*QGnc-7B-7984*	*TraesCS7B02G406300*	Protein transport protein Sec61 subunit γ
			*TraesCS7B02G406400*	B3 domain-containing protein
GPC	1B	*QGpc-1B-3068*	*STRG_1B.000039*	-
GPC	3B	*QGpc-3B-7236*	*STRG_3B.000102*	-
GPC	5D	*QGpc-5D-7948*	*TraesCS5D02G459900*	Vesicle-associated membrane protein, putative
GPC	6B	*QGpc-6B-361*	*TraesCS6B02G012600*	F-box family protein
GPC	7A	*QGpc-7A-11455*	*TraesCS7A02G476700*	Receptor-kinase, putative
GPC	7A	*QGpc-7A-6017*	*TraesCS7A02G246100LC*	Glycine-rich family protein
GKC	1D	*QGkc-1D-950*	*TraesCS1D02G024700*	Dehydration-induced 19-like protein
GKC	2B	*QGkc-2B-16317*	*TraesCS2B02G564200*	Heme oxygenase 1
GKC	6D	*QGkc-6D-3045*	*TraesCS6D02G115000*	Receptor kinase
StNC	1A	*QStnc-1A-5191*	*TraesCS1A02G238400LC*	E3 ubiquitin-protein ligase ORTHRUS 2
StNC	1D	*QStnc-1D-4837*	*STRG_1D.000058*	-
StNC	2A	*QStnc-2A-6246*	*TraesCS2A02G213508*	Pentatricopeptide repeat-containing protein, putative
StNC	2D	*QStnc-2D-2351*	*TraesCS2D02G124400*	F-box family protein
StNC	4B	*QStnc-4B-629*	*TraesCS4B02G024900LC*	P-loop containing nucleoside triphosphate hydrolases superfamily protein
			*TraesTL4B02G001100*	-
StPC	1B	*QStpc-1B-15303*	*TraesCS1B02G741400LC*	hAT transposon superfamily
StPC	2B	*QStpc-2B-4694*	*TraesCS2B02G131100*	Aspartic proteinase nepenthesin-1
StKC	4D	*QStkc-4D-1882*	*TraesCS4D02G229500*	Transposon Ty3-I Gag-Pol polyprotein
StKC	6A	*QStkc-6A-2415*	*TraesCS6A02G063300*	Kinase family protein
StKC	7D	*QStkc-7D-10397*	*TraesCS7D02G715100LC*	F-box protein
			*STRG_7D.000171*	-
			*TraesCS7D02G532100*	Cytochrome P450 family protein, expressed
GNA	1A	*QGna-1A-3190*	*STRG_1A.000030*	-
GPA	4A	*QGpa-4A-11638*	*STRG_4A.000155*	-
GKA	3D	*QGka-3D-4973*	*TraesCS3D02G549900LC*	Signal recognition particle protein
GKA	6A	*QGka-6A-6838*	*TraesTL6A02G007600*	-
GKA	7B	*QGka-7B-2040*	*TraesCS7B02G163800LC*	Spindle assembly checkpoint component
StNA	2A	*QStna-2A-18271*	*STRG_2A.000389*	-
			*TraesTL2A02G020500*	-
StNA	5D	*QStna-5D-4323*	*TraesCS5D02G295200*	F-box protein
StNA	7A	*QStna-7A-7244*	*STRG_7A.000092*	-
StPA	6A	*QStpa-6A-10015*	*TraesCS6A02G372600*	Phosphate regulon sensor protein PhoR
StPA	6A	*QStpa-6A-3769*	*TraesTL6A02G003400*	-
GNUE	2B	*QGnue-2B-3336*	*TraesCS2B02G106300LC*	Tetratricopeptide repeat (TPR)-like superfamily protein
GPUE	1A	*QGpue-1A-5873*	*TraesTL1A02G004100*	-
GPUE		*QGpue-6B-5100*	*STRG_6B.000077*	-
GKUE	4D	*QGkue-4D-1722*	*TraesCS4D02G195400*	Phosphoinositide phosphatase family protein
GKUE	5A	*Gkue-5A-9619*	*TraesCS5A02G383800*	Heat shock transcription factor
StNUE	4B	*QStnue-4B-1036*	*TraesCS4B02G033800*	Lysine--tRNA ligase
			*TraesCS4B02G034500*	Lysine--tRNA ligase
StPUE	4A	*QStpue-4A-5558*	*TraesCS4A02G475500LC*	Ypt/Rab-GAP domain of gyp1p superfamily protein
StPUE	6B	*QStpue-6B-15642*	*TraesCS6B02G783500LC*	B3 domain-containing protein
			*TraesCS6B02G440300*	Protein DETOXIFICATION
StKUE	2B	*QStkue-2B-7220*	*STRG_2B.000095*	-
			*TraesTL2B02G007800*	-
StKUE	3B	*QStkue-3B-1051*	*TraesCS3B02G008600*	Receptor-like kinase
StKUE	5A	*QStkue-5A-12724*	*TraesCS5A02G555500*	Autophagy protein 5
StKUE	6B	*QStkue-6B-5408*	*TraesCS6B02G133900*	Calmodulin-binding family protein, putative, expressed
StKUE	6D	*QStkue-6D-4669*	*TraesCS6D02G214800*	Bifunctional uridylyltransferase/uridylyl-removing enzyme
GPA/GPC	5A	*QGpa/Gpc-5A-12924*	*TraesCS5A02G556400*	Protein kinase family protein
GPA/StPUE	2B	*QGpa/Stpue-2B-18259*	*TraesCS2B02G609800*	3-isopropylmalate dehydrogenase
GKA/StPUE	4B	*QGka/Stpue-4B-1222*	*TraesTL4B02G002000*	-
			*TraesCS4B02G047600*	Threonine dehydratase
			*TraesCS4B02G047700*	Dynamin
			*TraesCS4B02G047900*	NADH-ubiquinone oxidoreductase subunit
			*TraesCS4B02G053900LC*	DNA mismatch repair protein MutS, type 2
			*STRG_4B.000022*	-
			*STRG_4B.000023*	-
			*TraesCS4B02G054110LC*	Zinc finger CCCH domain protein
			*TraesTL4B02G002100*	-
			*TraesCS4B02G049300*	GPN-loop GTPase-like protein
			*TraesCS4B02G049400*	Protein POLLEN DEFECTIVE IN GUIDANCE 1
StKA/StKC	2A	*QStka/Stkc-2A-12963*	*TraesCS2A02G654300LC*	histone deacetylase 2
			*TraesCS2A02G654500LC*	phosphatidyl inositol monophosphate 5 kinase

## Data Availability

Not applicable.

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
