# Peer review of "QTL Mapping and Candidate Gene Identifying for N, P, and K Use Efficiency at the Maturity Stages in Wheat"

_genes, 2023, doi:10.3390/genes14061168_

Round 1

Reviewer 1 Report

The manuscript “QTL mapping and candidate gene identifying for N, P and K use efficiency at the maturity stages in wheat” contains interesting data on NPK accumulation and use efficiency in wheat.

Line 26: RefSeq v1.1. – it is not clear in abstract that you mean Chinese Spring reference genome v.1.1.

Line 79 number of RILs should be provided

Line 129 LOD value should be determined by permutation test and may be different for each trait

File with supplementary data is not accessible (cannot be opened)

Genetic map used for QTL analysis was published in Chinese only it is not easily accessible – and it would be recommended to provide more details in this paper.

Discussion should be extended – and the role of selected candidate genes should be connected biochemically with NPK accumulation and usage.

Author Response

May 23, 2023

Dear reviewer,

We are very grateful to the reviewers for these important points and suggestions, which have greatly improved our manuscript. We have carefully revised our manuscript based on your suggestions to improve the quality of the manuscript. We sincerely hope that you will be satisfied with our responses and revisions.

  1. Q: Line 26: RefSeq v1.1. – it is not clear in abstract that you mean Chinese Spring reference genome v.1.1.

A: We have changed ‘RefSeq v1.1’ to ‘Chinese Spring (CS) RefSeq v1.1’. (lines 26-27)

  1. Line 79 number of RILs should be provided

A: We have changed ‘TL-RIL’ to ‘TL-RILs, 184 lines’. (line 81)

  1. Q: Line 129 LOD value should be determined by permutation test and may be different for each trait

A: In this study, the experiments were conducted in multiple environments and three different QTL software programs were used. It is difficult to detect high LOD values simultaneously in multiple environments. We think that the reliable QTLs should be detected under multiple environments at a properly LOD value instead of detected under single environment at a high LOD value. So we determined a relatively lower LOD value of 3.0. (lines 134-135)

  1. Q: File with supplementary data is not accessible (cannot be opened)

A: Supplementary data was submitted.

  1. Q: Genetic map used for QTL analysis was published in Chinese only it is not easily accessible – and it would be recommended to provide more details in this paper.

A: We have added more information in the text. ‘Each line of the TL-RILs was sequenced using RNA-Seq technology. The clean reads were mapped to the Chinese Spring (CS) RefSeq v1.1. A total of 31,445 polymorphic sub-unigenes were obtained, which represented 27,983 unigenes. The sub-unigenes were used to construct the UG-Map based on the physical positions.’ (lines 123-127)

  1. Q: Discussion should be extended – and the role of selected candidate genes should be connected biochemically with NPK accumulation and usage.

A: We have made large revise in the Discussion section. We added two candidate genes in Discussion, TraesCS6A02G372600 and TraesCS6B02G012600. (lines 242-255)

Thank you very much for your attention and time. Look forward to hearing from you.

Sincerely yours,

Si-shen Li

National Key Laboratory of Wheat Improvement, Shandong Agricultural University, Tai’an 271018, China

Tel.: +86 0538 8246503; Fax: +86 0538 8242226;

E-mail address: ssli@sdau.edu.cn

Reviewer 2 Report

This paper is pretty good,  it uses now standard methods and comes up with new information that others will follow up to advance our understanding of N P and K use by wheat crops. Below are some things of content and English that the authors might consider:

line 41, I do not think P is an "important component" of proteins, sugars and lipids, and the following 

table 1,  StNC line "Using", not "Uusing"

Discussion, from line 220. Given the amount of QTLs found for the traits, I think the authors should have considerable more discussion, especially of how their results compare/line up with the results of other researchers, and what is the possible relationship of the "candidate genes" and the mechanisms of NPK use in the plants.

Also, in consideration of the lines used in the RIL population and their pedigrees, I think some discussion of the possible prevalence of the  detected QTLs in current Chinese (and perhaps wider) wheat cultivars and breeding programs should be discussed.

This is OK.

Author Response

May 23, 2023

Dear reviewer,

We are very grateful to the reviewers for these important points and suggestions, which have greatly improved our manuscript. We have carefully revised our manuscript based on your suggestions to improve the quality of the manuscript. We sincerely hope that you will be satisfied with our responses and revisions.

  1. Q: line 41, I do not think P is an "important component" of proteins, sugars and lipids, and the following

A: We corrected the content. (line 41)

  1. table 1,  StNC line "Using", not "Uusing"

A: We have corrected. (Table 1, StNC line)

  1. Q: Discussion, from line 220. Given the amount of QTLs found for the traits, I think the authors should have considerable more discussion, especially of how their results compare/line up with the results of other researchers, and what is the possible relationship of the "candidate genes" and the mechanisms of NPK use in the plants.

A: We have made large revise to the Discussion section. We added ‘In addition, the QTLs and candidate genes in this study were identified according to the physical positions of the reference genome. It is difficult to accurately compare with QTLs obtained based on molecular markers.’ in the first paragraph of Discussion. (lines 230-232)

We added two candidate genes in Discussion, TraesCS6A02G372600 and TraesCS6B02G012600. (lines 242-255)

  1. Q: Also, in consideration of the lines used in the RIL population and their pedigrees, I think some discussion of the possible prevalence of the detected QTLs in current Chinese (and perhaps wider) wheat cultivars and breeding programs should be discussed.

A: We added ‘For TL-RILs, the male parent TN18 is derived from the cross ‘Laizhou137 × Yanong 19’ and the female parent LM6 is derived from the cross ‘924402 × Lvmai13’. The strain 924402 is one parent of cultivated variety ‘Jimai22’. Of these parents, Yannong19 and Jimai22 are the authorized varieties with great area and the core parents in wheat breeding programs of China. Moreover, the parent of TN18 is also a core parent in China. Hundreds authorized varieties were derive from Yannong19, Jimai22 and TN18. Therefore, the QTLs and their candidate genes in this study may be widely identified in the varieties of China.’ in Discussion section. (lines 261-267)

Thank you very much for your attention and time. Look forward to hearing from you.

Sincerely yours,

Si-shen Li

National Key Laboratory of Wheat Improvement, Shandong Agricultural University, Tai’an 271018, China

Tel.: +86 0538 8246503; Fax: +86 0538 8242226;

E-mail address: ssli@sdau.edu.cn
